# Cognitive Fusion Mediates the Relationship between Dispositional Mindfulness and Negative Affects: A Study in a Sample of Spanish Children and Adolescent School Students

**DOI:** 10.3390/ijerph16234687

**Published:** 2019-11-25

**Authors:** María García-Gómez, Joan Guerra, Víctor M. López-Ramos, José M. Mestre

**Affiliations:** 1Departamento de Psicobiología, Universidad de Murcia, 30001 Murcia, Spain; mgg99158@um.es; 2Departamento de Psicología, Universidad de Extremadura, 10071 Cáceres, Spain; joangb@unex.es (J.G.); vmlopez@unex.es (V.M.L.-R.); 3Instituto de Investigación y Desarrollo Social Sostenible (INDESS), Universidad de Cádiz, 11405 Jerez de la Frontera, Spain

**Keywords:** dispositional mindfulness, cognitive defusion, anxiety, mindfulness based on interventions, mental health, experiential avoidance, children and adolescents

## Abstract

Nowadays, mindfulness-based interventions (MBI) have experienced a remarkable development of studies among childhood and adolescent interventions. For this reason, dispositional mindfulness (DM) measures for children and adolescents have been developed to determine the effectiveness of MBI at this age stage. However, little is known about how key elements of DM (for example, cognitive de/fusion or experiential avoidance that both confirm psychological inflexibility) are involved in the mechanisms of the children and adolescents’ mental health outcomes. This research examined the mediating effect of cognitive fusion between DM and anxiety and other negative emotional states in a sample of 318 Spanish primary-school students (aged between 8 and 16 years, *M* = 11.24, *SD* = 2.19, 50.8% males). Participants completed the AFQ-Y (Avoidance and Fusion Questionnaire for youth), which is a measure of psychological inflexibility that encompasses cognitive defusion and experiential avoidance; CAMM (DM for children and adolescents), PANAS-N (positive and negative affect measure for children, Spanish version of PANASC), and STAIC (an anxiety measure for children). The study accomplished ethical standards. As MBI relevant literature has suggested, cognitive defusion was a significant mediator between DM and symptoms of both negative emotions and anxiety in children and adolescents. However, experiential avoidance did not show any significant mediating relationship. Probably, an improvement of the assessment of experiential avoidance is needed. MBI programs for children and adolescents may include more activities for reducing effects of the cognitive defusion on their emotional distress.

## 1. Introduction

Activities for children and adolescents based on mindfulness interventions are being used more and more by researchers, instructors, and therapists. Recent reviews have reported that mindful-based interventions (MBI) are feasible for young participants [1]. MBI for children and adolescents are promising initiatives, as there is evidence of their usefulness in treating childhood anxiety [2], reducing symptoms of obsessive-compulsive disorder [3], and their pathological concerns [4].

One the one hand, the most sought-after goals of MBIs and Acceptance and Commitment Therapy (ACT) interventions for children and adolescents is the promotion of psychological flexibility [5]. Psychological flexibility (PF) “is about being aware of thoughts and feelings that unfold in the present moment without needless defense, and depending on what the situation affords, persisting or changing behavior to pursue central interests and goals” (p. 868, [6]). For this reason, and from ACT assumptions, one way to increase psychological flexibility of individuals could be to promote producing better mental health outcomes [7,8]. Therefore, psychological inflexibility (PI) would be the opposite pole of the same dimension and, consequently, would have the opposite outcomes to those reported above [9,10].

On the other hand, dispositional mindfulness (DM) has been defined as a trait, in which individuals differ in how they accept their current contingencies and live with a commitment [11]. Therefore, DM is a predisposition or trait for living in a mindfulness way [12]. Considering any psychological intervention as a mechanism of change (for example, MBIs), DM as a trait might be included as a mechanism as well, which means that people with a high DM trait are more likely to behave in their lives with acceptance, gratitude, and forgiveness [12]. 

However, little is known about how the combined nature of the relationship between PI and DM regarding their potential predictions of mental-wellbeing outcomes in children and adolescents. For instance, some studies stated that the mechanisms through which being mindful can influence positive outcomes have only recently been explored; hence, it is necessary to explore these mechanisms [13]. Despite the fact that there are studies on how DM and/or PI are related to mental health outcomes, there are no studies on whether PI can play a mediating role between DM and emotional-wellbeing criteria. Moreover, there are no studies on what aspect of PI (cognitive fusion and experiential avoidance) may have a mediating effect on some indicators of mental health criteria. Therefore, there is a shortage of studies on PI and DM in children and adolescents [14,15].

Given the relevance of PI, the Avoidance and Fusion Questionnaire for Youth (AFQ-Y) was developed to assess this construct [16]. The instrument measures the two main interrelated processes which produce psychological inflexibility: cognitive fusion (CF) and experiential avoidance (EA) [16]. Cognitive fusion means responding as if private events were true, and thus, increase the tendency to avoid the internal experience, the opposite of a mindful attitude, such as present moment by moment awareness [17,18].

There are some promising antecedents with samples of children and adolescents using this PI measure. A longitudinal study with adolescents, experiential acceptance (the opposite of avoidance) predicted increasing positive affect and decreasing fear and sadness [14]. In a Dutch sample, PI (using a shorter version of the AFQ-Y and only one factor) and anxiety symptoms were positively correlated [9]. Using a sample of Spanish, the total score of AFQ-Y was related negatively to depression symptoms and positively to satisfaction with life [16]. In this last study, however, CF predicted depression and satisfaction with life, while experiential avoidance did not.

Although the AFQ-Y has been validated in various languages—Spanish [16], Italian [19], Swedish [20], and Dutch [9]—we found some controversies concerning the factor structure of the measure. Some authors described a single loaded factor (just psychological flexibility), while others found two factors (cognitive fusion and experiential avoidance). It is important to try to clarify this debate because it has consequences for the interpretation of the AFQ-Y. We believe that both cognitive fusion and experiential avoidance might have different natures for the same construct: psychological inflexibility (PI). Therefore, potentially, there could be different mediating effects of CF and EA between DM and negative/positive emotional outcomes.

A brief review of both AFQ-Y factors (cognitive fusion and experiential avoidance) helps to understand our point of view described above. Psychological flexibility (using an adaptive-positive term) has been defined as the ability to contact with present-moment awareness, and adjust behavior to valued aims and interests [21]; hence, PI is a lack of this ability. Psychological flexibility is a dynamic process which could be observed when a person: (1) adapts to a changing situational request, (2) redefines mental resources, (3) transforms perspective, and (4) balances needs, competing desires, and life domains [22]. For this reason, a lack of this ability, called psychological inflexibility, has been associated with different diagnostic categories [22,23,24] and is a predictor of emotional distress, anxiety, and attenuation of positive life appraisals and emotions [25].

Moreover, two processes have been included as factors of this psychological inflexibility topic: cognitive fusion and experiential avoidance. On the one hand, cognitive fusion (CF) is a process by which the individual becomes entangled with memories, thoughts, judgments, and evaluations and adjust behavior to the internal experiences [20]. Therefore, CF should have a mediating role between DM and some psychological outcomes [6]. On the other hand, experiential avoidance (EA) has been defined as a set of behaviors in which a person refuses to remain in contact with the internal experience such as thoughts, feelings, images, and sensations, labeling them as negative events [24,26]. Experiential avoidance can be considered adaptive if it is used as self-control strategies in the short term [27]. Hence, EA becomes problematic if used continuously and rigidly. According to this last point of view, EA has been related to psychological symptoms and problem behaviors [23] as well as anxiety, depression, emotional distress [26,27], and obsessive-compulsive symptoms [28].

At this point, it is necessary to understand how each of the two components of psychological flexibility contributes to the effects of DM on key criteria in the emotional wellbeing of children and adolescents. According to the antecedents previously reported, DM and psychological flexibility have significant relationships with positive and negative emotions, especially with both state and trait anxiety [28].

Regarding potential mediating effects of PI and its elements, it has been verified that psychological inflexibility fully mediated the effect of early maladaptive schemes on psychopathology in a non-clinical undergraduate sample [29]. Cognitive fusion had a mediating association with catastrophizing coping strategy in a chronic-pain community sample of young people [30]. Regarding experiential avoidance, some authors have suggested that it is important to understand models of psychopathology by assessing the role of experiential avoidance as a mediator and/or moderator [31,32]. Beyond these studies, little has been explored about the mediating effect of psychological flexibility, cognitive fusion, or experiential avoidance between DM and some positive or negative emotional states, especially in children and adolescents.

This cross-sectional study was designed to understand better how DM and PI are present in the mechanisms of emotional wellbeing among children and adolescents. We hypothesized that cognitive fusion and experiential avoidance (as components of the psychological inflexibility) have a mediating role between DM and emotional health criteria (positive/negative affectivity and both state and trait anxiety). 

## 2. Materials and Methods

### 2.1. Participants

Using a simple random sampling strategy, the sample had a total amount of 327 children and adolescents from nine primary and secondary schools in southern Spain. The schools, classroom and students were randomly selected from a huge area of the south of Spain. The age of participants ranged from 8 to 16 years old (*M* = 11.24, *SD* = 2.19, 50.8% males). Study information and consent forms were distributed to all parents (according to the Spanish Organic Law of Data Protection). Students were asked to fill out the questionnaires in one session during the class by supervising psychologists (M. G-G and J.G. authors of this article). Nine children were excluded due to errors or omissions in their responses or because their parents did not provide informed consent. 

In order to test the homogeneity of the sample, we conducted the Test of Levine for testing the homogeneity of the sample. Using *gender* (0 = male; 1: female) and *age* (0 = age < 12 years old (35.65% male and 36.28% female students), and 1 ≥ 12 years old (15.14% male and 12.93 female students), N = 317), Levine’s test showed that samples was homogenous by age and gender (F = 0.05, *p* = 0.82). 

In conformance with the Research and Ethical Commission of INDESS (University Research Institute on Social and Sustainable Development, University of Cadiz, Spain), we adhered to the following ethical recommendations: (a) all participants had to bring an informed consent from their parents, particularly those under 14 years old; (b) we had to inform and receive permission from every single Parents’ Association, and (c) the study had to be approved by an external ethical board (Ethical board of University of Murcia, Spain). Consequently, this research is subject to compliance with ethical standards: the 1964-Helsinki declaration and its later amendments or comparable ethical standards and. According to article 13.1 of the Spanish Organic Law of Data Protection, the "data of persons over fourteen years of age may be processed with their consent, except in those cases in which the Law requires the assistance of the holders of parental authority or guardianship. In the case of minors under 14 years of age, the consent of the parents or guardians will be required".

### 2.2. Measures

#### 2.2.1. Psychological Inflexibility: Cognitive Fusion and Experiential Avoidance

Psychological inflexibility (PI) was measured using the Spanish adaptation [16] of Avoidance and Fusion Questionnaire for Youth (AFQ-Y, [15]). This tool was developed to measure psychological inflexibility, and its components—cognitive fusion and experiential avoidance—in children and adolescents. It was adapted from the Acceptance and Action Questionnaire [33], which is a self-report measure for psychological inflexibility in adults. The AFQ-Y consists of 17 items that can be rated on a Likert scale ranging from 0 (not at all true) to 4 (very true). Example items include "I am afraid of my feelings" and "the bad things I think about myself must be true". Total scores ranged from 0 to 68. High scores of AFQ-Y indicate higher psychological inflexibility, experiential avoidance and cognitive fusion. The 2008-study of Greco et al. of AFQ-Y showed high reliability (α = 0.9) [15]. AFQ-Y total, cognitive fusion and experiential avoidance were promediated (ranged from 0 to 4) due to every single subscale have a different number of items. 

#### 2.2.2. Dispositional Mindfulness

To assess the DM of the participants, we used the validated Spanish version [34] of the Child and Adolescent Mindfulness Measure (CAMM; [35]). The CAMM consists of 10 items, answered on a five-point Likert scale, ranging from 0 (never true) to 4 (always true). An example of a CAMM item was ‘I keep myself busy so I don’t notice my thoughts or feelings ‘. However, for the Ten-Spanish CAMM version, items 5 and 10 had to be erased due to cultural biases among Spanish-children samples. Spanish CAMM-8 had better internal consistency and reliabilities than CAMM-10. Previous research using CAMM-8 has been widely reported in a pilot study with gifted children [36]. Recently, another Spanish validation of the CAMM in a larger sample of Chilean and Spanish children adolescents with similar outcomes regarding items 5 and 10; however, the authors recommended a shorter version of a seven-item CAMM (removing # 2, 5, and 10 from Greco’s original CAMM) [37]. Nonetheless, we used the total score of Spanish CAMM-8, which showed good levels of internal consistency (α = 0.82) [34] and its scoring was promediated (ranging from 0 to 4) for this study. The DM measure was also recoded into positive total scores, thus, higher scores reflected better dispositional mindfulness rather than the opposite. 

#### 2.2.3. Positive and Negative Affect

The Positive and Negative Affect Schedule for Children (Spanish validation ‘PANASN’, [38]). This measure was based on the original version of the instrument called PANAS-C [39]. This tool consists of a 30-item measure for children and young adolescents, which assesses several positive affects (PA; e.g., joyful) and negative affects (NA; e.g., lonely) with 15 items in each subscale. Moreover, it returned a score on Balance (PA–NA). Through a three-point Likert scale, ranging from 1 (slightly or never) to 3 (much), participants valued how they felt over the last few weeks. The minimum score for each scale is 15 and the maximum score is 45. Internal consistency, convergent and discriminant validity for the Spanish PANAS-C (named as PANASN) had been previously obtained [38]. This research used an average score of PANASN (from 1 to 3). We used the acronym PANASN for this study. Positive affect (PA), negative affect (NA), and balance were used as criteria. Original validation of PANASN showed a good internal consistency from 0.87 to 0.91 [38].

#### 2.2.4. Anxiety

The State-Trait Anxiety Inventory for Children (STAIC; [40]). This self-report scale has been frequently used to measure state and trait anxiety in children and adolescents. The instrument was created to measure the propensity to anxiety both as a transient state and as a relatively stable individual difference. To measure trait and state anxiety, this questionnaire incorporates two separate 20-item self-report rating scales. Spanish STAIC version has showed good internal consistency from 0.82 to 0.89 [40]. The participant has to evaluate the degree to which he or she experiences a particular symptom (e.g., I feel 1—not scared, 2—scared, and 3—very scared). By adding up the scores for each scale, the state-anxiety and trait-anxiety are obtained. Total scores for situational and baseline questions range, separately, from 20 to 60, with higher scores denoting higher levels of anxiety. Due to the special computation of the STAIC’s scoring, this measure was not promediated. 

#### 2.2.5. Data Analysis

Data analyses were conducted using SPSS 24.0 (IBM, Armonk, NY, USA) to describe and show the descriptive, intercorrelations, and reliabilities statistics. We used *M* (mean) and *SD* (standard deviation), Pearson’s correlations, and Cronbach’s internal consistency. 

Mediating analyses was checked and tested using the PROCESS (a plugging to SPSS), developed by Andrew F. Hayes, which does the centering and interaction terms automatically. However, prior to conduct mediating analyses with process, it is necessary to dummy code categorical variables with more than two categories before including them in the model.

## 3. Results

Before studying the mediation effects of psychological flexibility and its components, we show the results of the descriptive statistics, reliabilities, and correlations of all variables in Table 1. 

Table 1 shows interesting figures in order to be related to further mediating analyses. However, the primary goal was to determine the mediating roles of the two PI factors (experiential avoidance and cognitive fusion). Regarding *experiential avoidance,* despite negative and moderate significance relationships with negative affect, trait, and state anxiety, EA did not show significant influences on the different mediation analysis (negative affect, anxiety trait, and anxiety state); neither did age significantly covariate in such mediation analyses. Likewise, *gender* had no moderating role in further analyses. However, we did find a mediating role of *cognitive fusion* between DM and negative affect and both state and trait anxiety. We decided not to report mediation analysis with a balance criterion due to it being the result of the subtracted difference between the positive affection and the negative affection. This subtraction does not reflect a balance, but the difference between the totals of the positive and negative items of the PANASN.

Regarding DM, CF, and negative affect of PANASN mediation analysis, there was a significant indirect effect of DM on the Negative Affect through Cognitive Fusion, *b* = 0.051, 95% BC CI (Confidence Interval) [0.019, 0.086], as shown in Figure 1.

The confidence interval for the indirect effect is a bootstrapped CI based on 5000 samples. Similar findings appeared regarding anxiety state (STAIC). There was a significant indirect effect of DM on anxiety state through Cognitive Fusion, *b* = 0.83, 95% CI [0.24, 1.44], as shown in Figure 2.

The confidence interval for the indirect effect is a bootstrapped CI based on 5000 samples. Finally, there was also a significant indirect effect of DM on anxiety trait trough Cognitive Fusion, *b* = 0.84, 95% CI [0.29, 1.43], as shown on Figure 3.

The confidence interval for the indirect effect is a bootstrapped CI based on 5000 samples.

## 4. Discussion

The purpose of the present study was to assess the potential mediating role of cognitive fusion and experiential avoidance between DM and emotional wellbeing criteria such as negative affect and anxiety. However, our arguments should be considered tentatively for two main reasons. 

On one hand, although the benefits of MBIs for school-aged children have been reported (see three systematic reviews: [1,41,42]), most children-and-adolescents MBI studies use DM measures rather than a systematic observation of how mindful mechanisms work for better mental and emotional health outcomes after MBIs. Little is known about mindful mechanisms; thus, any mediation analysis could provide tentative but not, as yet, confirmatory ideas. In our study, we did not conduct an MBI; just a correlational study. However, at earlier stages of mediation analysis, these types of correlational studies serve useful purposes [43,44,45]. Furthermore, we used a DM measure, the Spanish eight-item CAMM [46]. We recoded the total score for a better understanding since CAMM assesses indispositional mindfulness rather than DM. The original CAMM [35] seems to be cultural biased when is translated and validated in Spanish samples [46,47]; despite good reliabilities with the 10-item CAMM, it fits better when some items were removed (# 5 and 10 for Turanzas’s version [34]; # 2, 5, and 10 for the García-Rubio et al. version [46]; and # 2, 3, 5, 6, and 10 for the Guerra et al. version [47]). Yet, our arguments about the predictor variable are tentative because DM shows a trait [12] rather than a well-trained set of thoughts and behaviors based on the principles of mindfulness training. 

This study used Spanish AFQ-Y [16]. This measure provided our two potential mediators (CF and EA) between DM and emotional criteria. Our findings regarding cognitive fusion mediating role were consistent. However, our concerns were addressed to experiential avoidance assessment issues. We have found some articles that report the role of the construct of experiential avoidance used by clinical researchers [39,48]. A systematic review of the EA has revealed its relationship with various psychopathological symptoms [31], this review sustains that EA is a factor in the etiology of maladaptive behaviors related to some mental disorders (e.g., generalized anxiety disorder). Although the current literature suggests that EA may be involved in various forms of psychopathology, some authors also alert to a "lack of theoretical integration and refinement concerning operationalizing and assessing experiential avoidance" [39]. Therefore, in the assessment of the construct for adults, adolescents, and children doubts remain as to whether AFQ-Y correctly measures the experiential avoidance construct. We believe that EA is a negative reinforcement process [49]. Avoid and escape are two key verbs related to understanding this process [50] and its relationship with psychopathologic symptoms [22]. In targeting experiential avoidance behavior, the central premise of behavioral theory and activation treatments is that increasing avoidance and escape behavior will decrease exposure to positive reinforcement for healthy outcomes (e.g., increasing depression symptoms, see [50]). The findings partially support the arguments that the role of negative reinforcement processes (like EA) in mental health (for instance, depression) depends more on temporality terms than on the certainty or not of avoidance-or-escape behaviors [51]. Therefore, we cannot confirm whether EA has a mediating role or not due to assessment issues with AFQ-Y regarding this construct (involved Spanish AFQ items are 6 to 9, 11, 12, 14, 15, and 17). Instead of responding about the degree of certainty in an EA item (‘I push away thoughts and feelings that I don’t like’), we suggest including items some key information about the temporality of such information in these EA items (‘I usually push away thoughts and feelings that I don’t like’). 

With these caveats in mind, Table 1 reports high moderate and significant relationships between DM and AFQ-Y factors—CF and EA— (from *r* = 0.57 to *r* = 0.93, all *p* < 0.001), and moderate relationships between DM, EA, and CF and negative affect and anxiety trait, and a low moderate relationship with anxiety state. There are potential casualties between a positive DM, low CF, and EA regarding negative affectivity and anxiety. Reliabilities showed a good index for all measures (predictor, mediators, and criteria). Age and sex variables had no relevant effects on the measures implemented in this study. However, only cognitive fusion seems to be a mediating role when we performed mediation and moderation analyses.

According to Kazdin’s earlier mediation analysis [52], we fulfilled the four relations conditions for a mediation analysis (basically, all involved variables are related to each other, and by controlling the mediator predictor-criterion, the relationship has to change). 

Cognitive fusion was positively correlated with anxiety and depression. Other studies have found a moderate association between CF and anxiety that was not due to negative affect; moreover, this association was independent of EA [53,54]. Furthermore, a study [55] examined the mediating effect of CF in a cross-sectional design. Authors measured the negative affect, depressive symptoms, rumination, cognitive fusion, and mindfulness in depressed outpatients and normative individuals. Their model showed that CF was the only significant mediator of the relationship between negative affect and depressive symptoms (accounting for 61% of the total variance) [55]. Moreover, a longitudinal study found that CF predicted and mediated the effect of mindfulness on negative affect, depression and posttraumatic stress symptoms [56]. Our results are in line with these studies, which highlighted the potential mediator role of CF in mental health. 

Some authors hold that mindfulness and cognitive defusion could be overlap constructs. Thinking about this relationship, it could be that when an individual pays attention to his thoughts as mental events, this increases the separation or defusion of his thoughts, seeing them with a perspective [57]. In fact, in some exercise of the ACT model, mindfulness is used to increased cognitive defusion [5]. Although both terms are related, mindfulness is a concept that encompasses aspects beyond cognitive fusion. However, studies seeking to address the mechanism of mindfulness and mental health find inconsistencies [58]. Previous literature suggested that the relation between mindfulness and mental health outcomes could be a function of the effect of mindfulness on cognitive fusion [17,59]. In our study, we found a strong relationship between both concepts and the mediator effect of CF on negative affect and anxiety. One possible explanation of our model is that CF could be proximal processes underlying the effect of mindfulness on mental health [60]. The non-judge perspective towards thought, being aware of them, without reaction, allows individuals to achieve a psychological distance from their thoughts [61]. These individuals change their old pattern of interaction with their thoughts. Consequently, the soaring effects from the sad mood to severe affective disorders could be broken [62]. Thus, the defuse perspective, which is given by mindfulness practice, may contribute to less dysregulated emotional outcomes. In conclusion, the cognitive aspect of DM could be the key factor in the contribution of mindfulness to mental health. This finding has an important impact on treatments to improve mental health. Interventions should take into account the main role of cognitive fusion in the promotion of mental health through mindfulness approaches. As such, this study underlines the role of mindfulness and helps to understand how mindfulness is beneficial to mental health and the mediator role of CF.

As for the relationship between mindfulness and EA, although research has documented the role of EA in anxiety [63], our study does not find support for the important role of EA in psychopathology, probably because AFQ-Y does not encompass the Experiential Avoidance construct adequately. However, another study found that EA was not a predictor of depressive symptoms and negative-emotions states, being the DM a better protector factor in depression than EA [62]. These results are consistent with our findings and suggest that avoidance behavior (towards internal or external events) it is not the main factor contributing to negative outcomes, rather it is cognitive aspects that seem to play a mediating role between DM and negative affect.

## 5. Conclusions

Despite some important limitations regarding this study (it is not a longitudinal study, no MBIs were performed, and size of the sample), there are some interesting readings to bear in mind to educational and clinical future implications. For example, activities to increase cognitive defusion (rather than cognitive fusion) may help to the students to satisfactory cope with internal and interpersonal situations, which involve negative emotional states such as anxiety. According to several authors with a huge contrasted MBI’s experience, cognitive defusion is watching at thoughts rather than from thoughts, noticing them instead of being catch from thoughts, trying to flow with thoughts and not to hold them [1,2,6,8,10,14,18,25]. However, regarding experiential avoidance is a topic soundly consistent to understand relevant mental mechanisms [20,24]; however, its measure clearly needs further improvements and a better criterion validity using long-term and short-term avoidance experiences.

## Figures and Tables

**Figure 1 ijerph-16-04687-f001:**
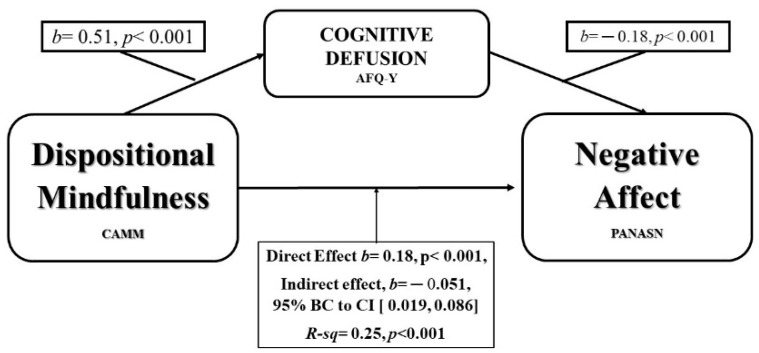
Model of DM as predictor of Negative Affect, mediated by Cognitive Fusion. AFQ-Y: Acceptation and Fusion Questionnaire for Youth.

**Figure 2 ijerph-16-04687-f002:**
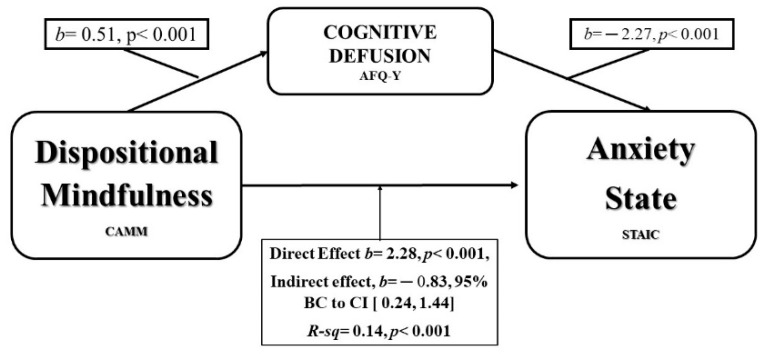
Model of Mindfulness as predictor of Anxiety-estate, mediated by Cognitive Fusion.

**Figure 3 ijerph-16-04687-f003:**
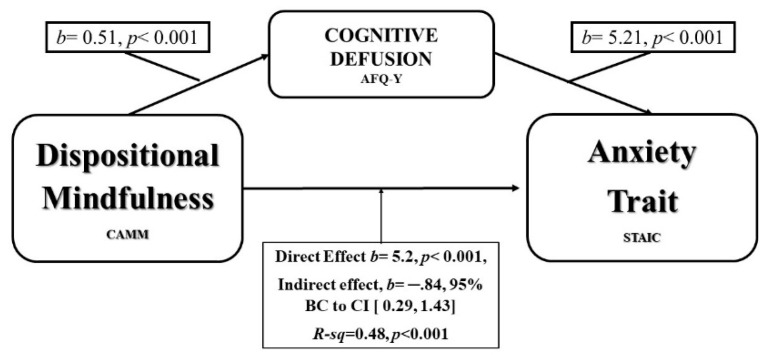
Model of DM as predictor of anxiety trait, mediated by Cognitive Fusion.

**Table 1 ijerph-16-04687-t001:** Descriptive and intercorrelations values for the empirical study (*N* = 318, 49.1% females). Cronbach reliabilities (α) for the study is also reported.

Variables	*Min—Max*	*M* (*SD*)	α	1	2	3	4	5	6	7	8	9	10
1 Age	8–16	11.25 (2.2)	-	*									
2 Gender	(0: male, 1: female)		-	−0.02	*								
3 CF_AFQ_Y	0–3.38	1.03 (0.70)	0.65	−0.12 *	0.02	*							
4 EA_AFQ_Y	0–4	1.87 (0.89)	0.76	−0.36 **	0.00	0.62 **	*						
5 AFQ_Y_Total	0–3.53	1.47 (0.72)	0.82	−0.29 **	0.01	0.87 **	0.93**	*					
6 DM_CAMM	0.4–4	2.66 (0.27)	0.73	0.23 **	0.00	−0.55 **	−0.60 **	−0.64 **	*				
7 PA_PANASN	1–3	2.35 (0.37)	0.72	−0.05	0.15 *	−0.07	−0.02	−0.04	−0.04	*			
8 NA_PANASN	1–2.6	1.61 (0.39)	0.78	−0.11 *	0.13 *	0.40 **	0.33 **	0.40 **	−0.47 **	−0.15 **	*		
9 BALANCE	−0.8–2	0.74 (0.58)	0.65	0.04	0.01	−0.31 **	−0.24 **	−0.30 **	0.29 **	0.75 **	−0.77 **	*	
10 ANX_S	20–55	29.97(6.41)	0.82	−0.09	−0.15 **	0.34 **	0.24 **	0.31 **	−0.31 **	−0.29 **	0.37 **	−0.44 **	*
11 ANX_T	20–54	33.37(7.19)	0.85	-0.18 *	0.07	0.50 **	0.49 **	0.55 **	−0.67 **	−0.00	0.62 **	−0.42 **	0.34 **

* *p* < 0.05; ** *p* < 0.01. 3 CF: Cognitive fusion; 4 Experiential avoidance; 5 Total Score of AFQ-Y (Average); 6 Dispositional Mindfulness (DM) of CAMM; 7 Positive Affect PANASN; 8 Negative Affect PANASN; 9 Balance (POS—NEG) PANASN; 10 ANX_S: Anxiety State STAI-C; and 11 ANX_T: Anxiety Trait STAI-C.

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
