# Peer review of "Cognitive Fusion Mediates the Relationship between Dispositional Mindfulness and Negative Affects: A Study in a Sample of Spanish Children and Adolescent School Students"

_ijerph, 2019, doi:10.3390/ijerph16234687_

Round 1
Reviewer 1 Report
This study develops an interesting topic but, there are considerable items that need the author’s' attention, as follows: Comment 1. Participants It is necessary to specify if the sample was randomly selected: the schools? the classrooms? the participants? In addition, I suggest to include the test of homogeneity to check if sample’s distribution across gender and age is homogenous. Comment 2. The authors have indicated that the age of the participants ranged between 8 and 16 years. Please, could you clarify if there are primary students who are 16 years old? Furthermore, it is not necessary to mention the people’s names who distribute the questionnaires. Comment 3. A section is missing. I recommend to add data analysis and the statistical program used in this study. Comment 4. Why does the score of the PANAS factors transformed and not the STAIC? Comment 5. Table 1 is explained in the discussion, but it should described in the results section and in the discussion stick to what corresponds in this section. Comment 6. Lack of clarity in the presentation of the results. Comment 7. Please, I suggest to review carefully the references, for instance: Reference 6: when referring to the pages it says 865-78. Reference 7: when referring to the pages it says 1041-56. Reference 26: when referring to the pages it says 208-16. Reference 36: when referring to the pages it says 606-14 Reference 45: incomplete.
Author Response
Thanks for reviewing and adding such remarkable improvements:
1.- we provided now further explanations regarding sampling and participants. We included in data analyses section info about the homogeneity of the sample.
2.- Yes, according to the age of the sample, it is not only primary-school students- There are also secondary students. The title was shortened and changed.
3.- A data analysis section was included
4.- STAIC was not promediated due to its computing score. It is not recommendable in this case.
5.- Table 1 was sticked
6.- references were fixed
Attached we upload the last version. We have also a track-change version in case you want it.

Reviewer 2 Report
The manuscript entitled “Cognitive fusion mediates the relationship between dispositional mindfulness and anxiety and other negative emotions: a study in a sample of Spanish primary-school students” is well-written and it is interesting. However, the topic has been widely studied during the last year, and it is not novel. Further, some changes should be done in the manuscript by authors previously to be considered for publication.
Title:
The title is too long and includes some confused terms. Which is the meaning of “other negative emotions”? Based on the obtained results, seems to be negative affect. Authors should consider the elaboration of a shorter title.
Introduction
Taking into account the nature of the journal, mainly oriented to the public health, more information about mindfulness trait should be included in this section. This fact would help readers to understand the results and the main conclusions of the study.
Along the introduction section, authors should avoid any personal reference as “we believe” or “our review of the literature”. This type of expressions go in detrimental of the scientific sound of the manuscript.
Materials and Methods
Authors stated that participants ranged from 8 to 16 years old. Are all of the students in primary education? The Spanish educative system entails that students start the secondary school at 11-12 years old. Authors should clarify this issue in detail, taking into account that could modify deeply the obtained results and sample characterization.
The study design has been evaluated by any official Ethical Committee previously to be conducted? The registration code of the Committee approval should be included.
Measures
The reliability coefficients of the employed evaluation instruments should be included, both the value of the original version and the obtained in the present sample.
Authors must include a section of "Data analysis" after the measures description.
Results
Again, authors should avoid personal references, as "we show"...
In mediation analyses, the explained variance in each model should be added.
Discussion
Authors start the discussion describing the limitation of the study. This information should be included at the end of this section.
A paragraph about strengths and limitations of the study should be included at the end of the discussion section.
Which are the educative and clinical implications of the study? It could be relevant to add some information in this regard.
Based on the obtained results, authors should propose what type of future research is needed in the studied topic.
Author Response
Thanks for the commentaries and recommendations
we followed all the suggestions and recommendations
1.- We shortened the title and fixed confusing terms
2.- introduction section: we included more info about cognitive fusion and erased all "we, our, us" sentences. However, APA 6.0 strongly recommends using these pronouns. We agree to change for a better scientist approaching.
3.- We included more info about participants, however, these ethical committees have not got codes yet.
4.- Measures: we included all the original reliabilities of the measures. In table 1 we included all the study's reliabilities.
5.- We included data analyses section
6.- The explained variance is set in the figures
7.- Discussion: we included the limitations at the end of this section and provided further implications for educational and clinical settings.
We have also got a track-change MS version if the reviewer wants it.

Round 2
Reviewer 1 Report
-The homogeneity information of the sample should be indicated in the participants section.
-Indicate the gender and age group distribution with the values of the Χ2 in the SPSS contingency table.
-In participants section the writtennexpresion "eight" by thw number 8.
Author Response
Thanks for the suggestions. We have included the homogeneity information in the participants' epigraph. Besides, we have included age and gender distribution information.
Regarding change "eight" by "8", according to APA 6.1.1 under 10, it should be written with letters. However, we agree to change it using the number "8".
Changes were yellow tracked in the MS

Reviewer 2 Report
Although authors have revised the manuscript and some changes have been implemented, some issues remain not solved in the manuscript:
The sample is composed by children and adolescents. Due to the inherent differential characteritics between the two type of participants, it could be necessary that authors conduct the analyses separately for children and adolescents. Or at least, separately for students of primary and secondary school. Due to the ethical conventions in scientific research, the ethical code should be included in the manuscript. The explained variance of each model should be included in each figure.Author Response
Thanks for the suggestions and recommendations,
Regarding the sample's age, we reported that "age" had no mediating effects. This is a cross-sectional study instead of a longitudinal one. Dividing the sample into two groups (changing a quantitative variable into a qualitative one) would not offer a better approach due to our goal was not related to the age of the sample or their development.
However, we did include the explained variance in every single figure of the mediating effects.
